# Sleep regularity and mortality: a prospective analysis in the UK Biobank

**Lachlan Cribb[1†], Ramon Sha[1†], Stephanie Yiallourou[1†], Natalie A Grima[1], Marina Cavuoto[1,2], Andree-Ann Baril[3‡], Matthew P Pase[1,4]\*‡**

[1]Turner Institute for Brain and Mental Health, School of Psychological Sciences, Monash University, Melbourne, Australia; [2]National Ageing Research Institute, Melbourne, Australia; [3]Douglas Mental Health University Institute, McGill University, Montreal, Canada; [4]Harvard T.H. Chan School of Public Health, Harvard University, Boston, United States

**\*For correspondence:**
matthew.pase@monash.edu

†Joint first authors

‡Joint senior authors

**Competing interest:** The authors declare that no competing interests exist.

## Abstract

**Background:** Irregular sleep-wake timing may cause circadian disruption leading to several chronic age-related diseases. We examined the relationship between sleep regularity and risk of all-cause, cardiovascular disease (CVD), and cancer mortality in 88,975 participants from the prospective UK Biobank cohort.

**Methods:** The sleep regularity index (SRI) was calculated as the probability of an individual being in the same state (asleep or awake) at any two time points 24 hr apart, averaged over 7 days of accelerometry (range 0–100, with 100 being perfectly regular). The SRI was related to the risk of mortality in time-to-event models.

**Results:** The mean sample age was 62 years (standard deviation [SD], 8), 56% were women, and the median SRI was 60 (SD, 10). There were 3010 deaths during a mean follow-up of 7.1 years. Following adjustments for demographic and clinical variables, we identified a non-linear relationship between the SRI and all-cause mortality hazard (p [global test of spline term]<0.001). Hazard ratios, relative to the median SRI, were 1.53 (95% confidence interval [CI]: 1.41, 1.66) for participants with SRI at the 5th percentile (SRI = 41) and 0.90 (95% CI: 0.81, 1.00) for those with SRI at the 95th percentile (SRI = 75), respectively. Findings for CVD mortality and cancer mortality followed a similar pattern.

**Conclusions:** Irregular sleep-wake patterns are associated with higher mortality risk.

**Funding:** National Health and Medical Research Council of Australia (GTN2009264; GTN1158384), National Institute on Aging (AG062531), Alzheimer's Association (2018-AARG-591358), and the Banting Fellowship Program (#454104).

## eLife assessment

This manuscript provides **fundamental** findings on the association between sleep regularity and mortality in the UK Biobank, which is a popular topic in recent sleep and circadian research in population-based studies. The study is based on a large accelerometer study with validated follow-up of incident diseases and deaths, and the data quality and large sample size are **convincing** and strengthen the credibility of the conclusion. This will be of wide interest to researchers in the sleep study field, epidemiologists, practicing clinicians and the general public.

## Introduction

Circadian rhythms are endogenous cycles in physiological, hormonal, and behavioral processes largely synchronized to the external 24 hr light-dark cycle. The sleep-wake cycle is perhaps the most notable biological process that follows the 24 hr circadian rhythm (*Zee and Vitiello, 2009*). The timing of light

exposure is the primary external driver of circadian rhythms. Therefore, rapid changes in sleep timing can cause circadian misalignment through fluctuating light-dark exposure (*Zeitzer et al., 2000*).

Circadian misalignment is associated with several age-related diseases, including cancer and cardiovascular disease (CVD) (*Torquati et al., 2018*; *International-Agency-for-Research-on-Cancer, 2010*; *Janszky and Ljung, 2008*). However, the health impacts of irregular sleep-wake timing are still emerging. This remains an important area of study since modern societal and lifestyle trends, including exposure to artificial and blue light at night, longer work hours, shift work, and the 24-7 lifestyle, have blurred the distinction between day and night, increasing the propensity for circadian disruption (*Walker et al., 2020*). The present study assessed the relationship between sleep regularity and the risk of incident all-cause mortality, cancer mortality, and CVD mortality in the UK Biobank (UKB). We measured sleep regularity via accelerometry to calculate the sleep regularity index (SRI), a new metric sensitive to differences in sleep-wake timing on a circadian timescale.

## Methods

### Participants

Over 500,000 adults aged 40–69 years were recruited to the UKB cohort between 2006 and 2010 across 22 assessment centers. Participants were invited by the UK National Health Service patient registers, resulting in a 5.5% participation rate. Respondents were more likely to be older, female, and less likely to live in socioeconomically deprived areas than the general population (*Fry et al., 2017*). Baseline demographics, medical history, lifestyle, vitals, and blood samples were collected. A total of 106,053 participants completed a 7-day wrist-worn accelerometer study through random selection between February 2013 and December 2015. UKB has approval from the North West Multi-centre Research Ethics Committee as a Research Tissue Bank (RTB) approval (No. 16/NW/0274; 21/NW/0157). This approval means that the present study operates under the RTB approval and a separate ethical clearance was not required.

### Measurement of sleep regularity

Accelerometry data were collected using a wrist-worn device (Axivity AX3, United Kingdom) over a 7-day/night period. Estimated sleep status (awake or asleep) at a given time was calculated using the open-source R package GGIR version 2.7-1 (*Migueles et al., 2019*), using available algorithms (*van Hees et al., 2015*; *van Hees et al., 2018*). To distinguish sleep from sustained periods of inactivity without reference to a sleep diary (not available in the UKB), GGIR uses an algorithm to determine a daily 'sleep period time window' for each participant (*van Hees et al., 2018*). This defines the time window between the onset and end of the main daily sleep period, during which periods of sustained inactivity are interpreted as sleep. The algorithm does not, by default, detect bouts of sleep outside of this window and hence is not able to identify naps. Accelerometry data of low quality were removed using established UKB criteria (Appendix 1). Most participants (88%) provided complete accelerometry data. Participants with fewer than two valid SRI measurements (i.e., less than 2 contiguous 24 hr wear periods; <1%) were excluded. In total, 88,975 (84%) participants provided valid SRI data and were included in the study.

The SRI captures the probability of a participant being in the same state (asleep or awake) at any two time points 24 hr apart (*Phillips et al., 2017*). An individual who sleeps and wakes at precisely the same time each day would have an index of 100, whereas an individual who sleeps and wakes at entirely random times would have an index of 0. Each participant provided $k$-1 SRI measurements (where $k$ is the number of valid 24 hr periods), one for each contiguous 2-day pair. These SRI measurements were averaged using a linear mixed effects model with a random intercept for the participant and fixed effects for the day of the week and daylight savings transition. The average SRI was standardized over the day of the week and daylight savings transitions, so all SRI results were comparable.

### Mortality ascertainment

Mortality occurrence was identified through linkage with NHS Digital for participants from England and Wales and the NHS Central Register for participants from Scotland, with complete records available until January 2022. Death records included the date of death and the ICD-10 code for the primary cause. ICD-10 codes I00-I99 and C00-C97 defined CVD and cancer mortality, respectively.

## Ascertainment of disease status at baseline

History of cancer (ICD-10 codes D00-D09 and D37-D48), diabetes (codes E10-E14), mental and behavioral disorders (codes F00-F99), nervous system disorders (codes G00-G99), and CVD (codes I00-I99) at the time of the accelerometry study were ascertained through self-report at the UKB baseline session and through linkage with hospital inpatient records using the above ICD-10 codes. Linkage with hospital inpatient records was also used to identify disease occurrence between the UKB baseline session and the time of the accelerometry study.

## Data analysis

Data analysis was performed using R version 4.2.1. Cox proportional hazards models were used to examine associations between the SRI and incident all-cause mortality, CVD mortality, and cancer mortality. Surveillance for mortality commenced from the time of accelerometry (2013–2015) until the end of follow-up (January 2022), with a median follow-up time of 7.1 years (Q1, Q3: 6.6, 7.6). Non-events were censored at the last date they were known to be alive. For CVD and cancer mortality, deaths from competing causes were censored at the time of death. The SRI and all continuous confounders were modeled with restricted cubic splines with knots at the 10th, 50th, and 90th percentiles to allow for departures from linearity. Effect modification was assessed by adding product terms to Cox models. Missing data were infrequent (<2%) for most confounder variables and were imputed (10 imputations) by predictive mean matching using the aregImpute function of R package *Hmisc* (*Harrell and Harrell, 2019*).

In addition to Cox models, discrete-time hazards models, including an interaction between SRI and time (aggregated into 3-month intervals and modeled with a restricted cubic spline with knots at the 5th, 35th, 65th, and 95th percentiles), were fitted to relax the assumption of proportionality and allow hazard ratios (HRs) to vary over time (*Singer and Willett, 2003*). The SRI by time interaction in this model provided a test of proportionality (a small p-value would indicate strong evidence against the proportional hazards assumption). Time-varying HRs were then displayed visually. In cases where HRs showed clear time variation (i.e., hazards were non-proportional), we nonetheless present HRs from the Cox models as these can be interpreted as a weighted average of the time-varying HRs (*Stensrud and Hernán, 2020*). The discrete-time hazards model for all-cause mortality was also used to estimate standardized cumulative incidence (risk) across levels of SRI, with confidence intervals obtained by bootstrapping (*Hernán and Robins, 2010*). To reduce computation demand, only single imputation was used for the discrete-time hazards models.

All models were adjusted for the following variables that were selected using a directed acyclic graph (Appendix 3—figure 1): age, sex, ethnicity (White, Asian, mixed race, Black, or other), Townsend deprivation index, retirement status (retired vs. all other work arrangements), shift work (shift worker vs. non-shift worker), sick or disabled (self-reported employment category), household income (ordinal with five levels), highest level of education (ordinal with six levels), smoking status (current, former, never), smoking (pack years), and use of sedative, antidepressant, or antipsychotic medication.

## Sensitivity analyses

We fitted a second statistical model to determine whether the observed associations were independent of sleep time and disruption. Therefore, Model 2 included additional adjustments for overnight sleep duration and wake after sleep onset (WASO), averaged across accelerometry days (plus primary model covariates). In the second sensitivity analysis (Model 3), we adjusted for history of several diseases at baseline (cancer, CVD, mental and behavioral disorders, nervous system disorders, diabetes), in addition to the variables in the primary model. These variables were included as part of a sensitivity analysis as it is unclear whether they may be mediators or confounders of the SRI-mortality relationship. Long-standing irregular sleep may lead to prevalent disease (or a history of disease) at baseline and influence disease risk factors (*Fritz et al., 2021*; *Huang and Redline, 2019*; *Huang et al., 2023*; *Zuraikat et al., 2020*), indicating that these disease variables may play a mediating role (and consequently should not be adjusted). Conversely, such diseases may have effects disruptive to regular sleep and these variables may therefore confound the SRI-mortality relationship. Disease risk factor variables body mass index (BMI), moderate and vigorous physical activity (accelerometry-derived), systolic blood pressure (BP), and use of BP lowering medication, in addition to the variables

in Model 3, were included in a final sensitivity analysis (Model 4), as it is similarly unclear whether they may confound or mediate the SRI-mortality relationship.

## Comparison of SRI with other regularity measures

Preliminary reports which identified irregular sleep as a potential CVD risk factor measured sleep regularity as the amount of deviation in sleep patterns from an individual's average (i.e., the standard deviation [SD] of nocturnal sleep duration and sleep onset time) (*Nikbakhtian et al., 2021*; *Huang et al., 2020*). To contrast these SD-based metrics with the SRI, we fitted independent Cox models (each with primary model covariates) and estimated HRs for all-cause mortality for each of the three measures. Additionally, we added the SRI to a model containing both SD-based regularity measures (alongside primary model covariates) to test whether the SRI contained additional mortality risk information beyond that captured by the two SD metrics.

## Results

*Table 1* displays sample characteristics. The final sample size was 88,975. There were 3010 all-cause deaths during a median follow-up of 7.1 years (Q1, Q3: 6.6, 7.6). The most common primary cause of death was cancer (*n*=1701, 57%) followed by CVD (*n*=616, 20%).

## SRI and all-cause mortality

We identified a non-linear association between the SRI and all-cause mortality hazard (*p* [global test of spline term]<0.001) (*Figure 1*). Compared to the sample median (SRI = 61), mortality rates were highest among those with the most irregular sleep and decreased almost linearly as SRI approached its median, after which the decrease began to plateau (*Figure 1*). HRs, relative to the median SRI, were 1.53 (95% CI: 1.41, 1.66) for participants with SRI at the 5th percentile (SRI = 41) and 0.90 (95% CI: 0.81, 1.00) for those with SRI at the 95th percentile (SRI = 75), respectively. Standardized cumulative incidence curves for all-cause mortality are displayed for the SRI at the 5th percentile, median, and 95th percentile in *Figure 2*. There was little indication that HRs varied according to age (*p* interaction = 0.48), sex (p=0.36), household income (p=0.62), sleep duration (p=0.47), moderate to vigorous physical activity (p=0.13), history of CVD (p=0.48), or cancer (p=0.29).

There was strong evidence against the proportionality assumption in the discrete-time hazards model (*p* [time × SRI interaction]<0.001). Time-varying HRs for the 5th and 95th SRI percentiles compared to the median are displayed in the appendix (Appendix 3—figure 2). For the 5th percentile relative to the median, HRs were greatest in the earliest period of follow-up (HRs >2), declining until approximately 2.5 years, after which they remained approximately stable with an HR of around 1.5. There was no clear time variation in the HR for the 95th percentile of SRI vs. the median.

## CVD-specific mortality

The SRI was associated with CVD-specific mortality in the primary model (*p* [global]<0.001; *Figure 1*). HRs, relative to the median SRI, were 1.88 (95% CI: 1.61, 2.21) and 0.93 (95% CI: 0.73, 1.20) for the 5th and 95th percentiles, respectively. There was no evidence of non-proportional hazards in the discrete-time hazards model (*p* [time × SRI interaction]=0.57). There was little indication that HRs varied according to age (*p* interaction = 0.17), household income (p=0.30), sleep duration (p=0.69), moderate to vigorous physical activity (p=0.95), history of CVD (p=0.16), or history of cancer (p=0.24). HRs for low SRI were larger for females than males (*p* interaction = 0.006; Appendix 3—figure 4).

## Cancer-specific mortality

The SRI was associated with cancer mortality in the primary model (*p* [global]<0.001). HRs, relative to the median SRI, were 1.36 (95% CI: 1.22, 1.53) and 0.89 (95% CI: 0.77, 1.02) for the 5th and 95th percentiles, respectively. There was strong evidence of non-proportional hazards in the cancer mortality discrete-time hazards model (*p* [time × SRI interaction]<0.001). HRs, for the 5th percentile vs. the median, were large at the beginning of follow-up (HRs >2) and declined until approximately 4 years, after which they were small (~1.05) and compatible with the null (Appendix 3—figure 3). There was no indication that HRs for the 95th SRI percentile relative to the median varied over follow-up. There was little indication that hazard ratios varied according to age (*p* interaction = 0.48),

**Table 1.** Sample characteristics (*n*=88,975).

| Characteristic | SRI tertile | | |
| --- | --- | --- | --- |
| | <56.8 n=29,361 | 56.8–65.2 n=29,361 | >65.2 n=30,252 |
| Sex (male), *n* (%) | 16,429 (56%) | 12,747 (43%) | 9691 (32%) |
| Age (years) | 62.3 (7.8) | 61.8 (7.8) | 61.6 (7.9) |
| BMI | 27.8 (4.9) | 26.6 (4.3) | 25.8 (4.1) |
| Ethnicity, *n* (%) | | | |
| Asian | 1199 (4.1%) | 1110 (3.8%) | 1114 (3.7%) |
| Black | 159 (0.5%) | 107 (0.4%) | 76 (0.3%) |
| Mixed race | 1024 (3.5%) | 831 (2.8%) | 658 (2.2%) |
| White | 26,621 (91%) | 27,070 (93%) | 28,210 (93%) |
| Other | 233 (0.8%) | 145 (0.5%) | 129 (0.4%) |
| Townsend deprivation index (score units) | −1.36 (3.01) | −1.79 (2.76) | −2.01 (2.63) |
| Household income* (thousands), n (%) | | | |
| <18 | 4917 (19%) | 3601 (14%) | 3266 (12%) |
| 18–30 | 6665 (25%) | 6225 (24%) | 6450 (24%) |
| 31–50 | 7352 (28%) | 7741 (29%) | 7843 (29%) |
| 51–100 | 5888 (22%) | 6816 (26%) | 7280 (27%) |
| >100 | 1633 (6.2%) | 1994 (7.6%) | 2159 (8.0%) |
| Retired, *n* (%) | 9467 (32%) | 8984 (31%) | 9442 (31%) |
| Shift worker, *n* (%) | 1869 (6.4%) | 1146 (3.9%) | 912 (3.0%) |
| Smoking status, *n* (%) | | | |
| Current | 2790 (9.5%) | 1863 (6.4%) | 1451 (4.8%) |
| Former | 10,956 (37%) | 10,587 (36%) | 10,367 (34%) |
| Never | 15,526 (53%) | 16,830 (57%) | 18,361 (61%) |
| Sedative medication, *n* (%) | 334 (1.1%) | 231 (0.8%) | 216 (0.7%) |
| Antidepressant medication, *n* (%) | 2217 (7.6%) | 1543 (5.3%) | 1385 (4.6%) |
| History of cancer, *n* (%) | 3941 (13%) | 3800 (13%) | 3897 (13%) |
| History of CVD, *n* (%) | 13,949 (48%) | 11,602 (40%) | 10,727 (35%) |
| History of diabetes, *n* (%) | 1976 (6.7%) | 1059 (3.6%) | 724 (2.4%) |
| History of neurological disease, *n* (%) | 3946 (13%) | 3380 (12%) | 3467 (11%) |
| History of mental/behavioral disorder, *n* (%) | 3521 (12%) | 2482 (8.5%) | 2184 (7.2%) |
| Average night time sleep duration (hours; actigraphy-derived) | 6.33 (0.97) | 6.59 (0.78) | 6.79 (0.66) |
| Average night time wake after sleep onset (hours; actigraphy-derived) | 0.86 (0.30) | 0.80 (0.26) | 0.70 (0.23) |
| Sleep duration SD, hours | 1.33 (0.64) | 1.11 (0.52) | 0.94 (0.48) |
| Sleep onset time SD, hours | 1.41 (1.12) | 0.96 (0.64) | 0.72 (0.51) |
| SRI, score units | 48.5 (7.3) | 61.2 (2.4) | 70.3 (3.7) |

Data are mean (SD), unless specified otherwise. *pounds. SRI = sleep regularity index; CVD = cardiovascular disease.

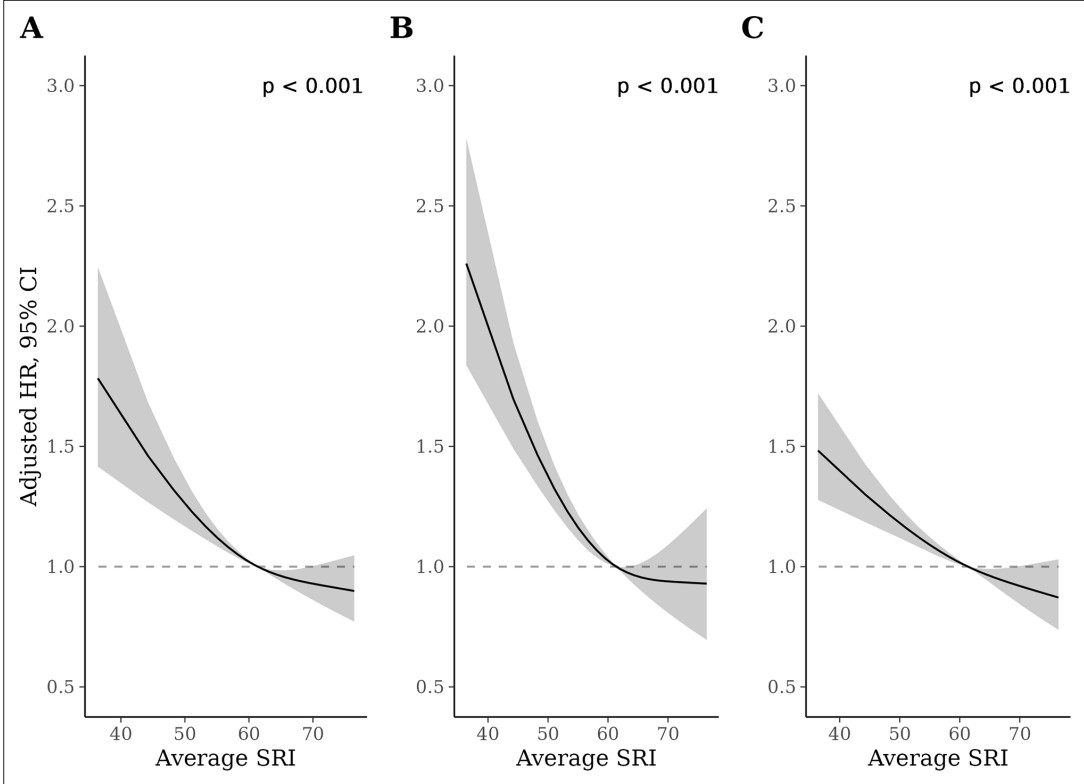

**Figure 1.** Adjusted hazard ratios (HRs) for all-cause (**A**), cardiovascular disease (CVD) (**B**), and cancer (**C**) mortality. p-Values from global (2 degrees of freedom) test of spline term. HRs are relative to the median sleep regularity index (SRI) (SRI = 60). HRs for all-cause mortality, CVD mortality, and cancer mortality were estimated using Cox proportional hazards models, adjusted for age, Townsend deprivation index, sex, antidepressant, antipsychotic, and sedative medication, ethnicity, household income, education, smoking status (former, current, never), smoking pack years, shift work, retirement status, and sick or disabled (self-reported employment category). All continuous confounders and the SRI were modeled with restricted cubic splines (knots at 10th, 50th, and 90th percentiles) to allow for departures from linearity.

sex (p=0.36), household income (p=0.62), sleep duration (p=0.49), moderate to vigorous physical activity (p=0.42), history of CVD (p=0.82), or history of cancer (p=0.12).

## Sensitivity analyses

Sensitivity analyses are displayed in the appendix (Appendix 3—figures 5–7). Overall, results were similar and not meaningfully altered following adjustments for sleep time and WASO (Model 2) or history of disease at baseline, including cancer and CVD (Model 3). The SRI remained associated with mortality after further adjustments for disease history, BMI, systolic BP, BP treatment, and physical activity (Model 4), though effect sizes were attenuated. For example, when comparing the 5th percentile to the median, HRs were 1.22 (95% CI: 1.07, 1.39) for all-cause, 1.43 (95% CI: 1.21, 1.69) for CVD, and 1.15 (95% CI: 1.01, 1.29) for cancer mortality.

## Comparison of SRI with sleep duration SD and sleep onset time SD

The SRI was modestly negatively correlated with the sleep duration SD (−0.32) and sleep onset time SD (−0.42; see correlation matrix in *Appendix 3—table 1*). *Figure 3* displays HRs, relative to the median, for the SRI, sleep duration SD, and sleep onset time SD. For each measure, greater sleep irregularity (i.e., lower SRI or higher SD representing more day-to-day variability) was associated with an increased all-cause mortality rate in independent models (all *p* [global]<0.001). HRs, for low regularity compared to the median, were largest for the SRI (*Figure 3*). The addition of the SRI to a model containing both SD metrics (alongside primary model covariates) improved model fit (*p* [likelihood ratio test]<0.001).

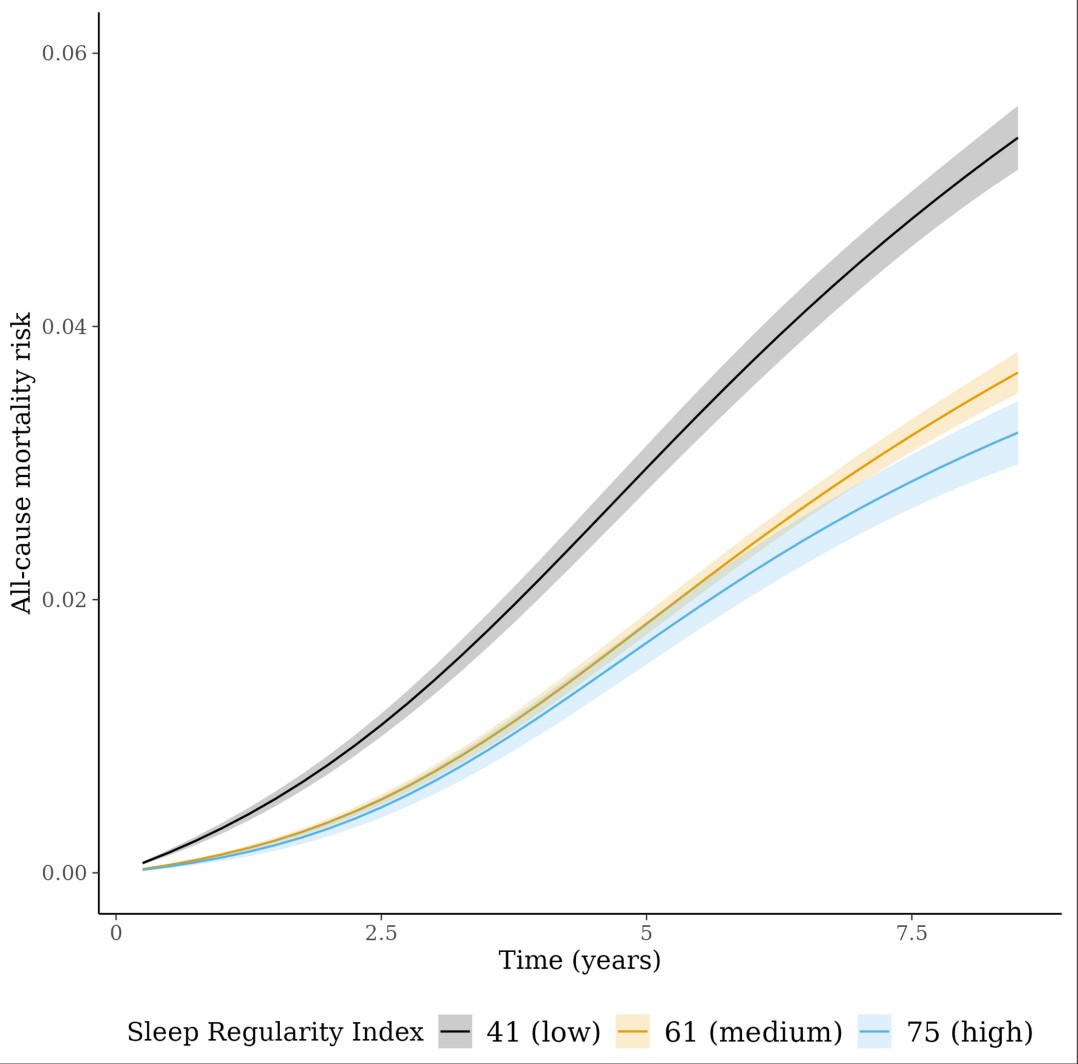

**Figure 2.** Cumulative incidence of all-cause mortality across sleep regularity index (SRI). Standardized cumulative incidence of all-cause mortality for SRI at 41 (5th percentile), 61 (median), and 75 (95th percentile). Estimates from a discrete-time hazards models including an interaction between SRI and time (aggregated into 3-month intervals and modeled with a restricted cubic spline with knots at the 5th, 35th, 65th, and 95th percentiles) and primary model covariates. Confidence intervals were obtained by bootstrapping.

Conversely, the addition of sleep duration SD and sleep onset time SD to a model containing the SRI (and primary model covariates) did not meaningfully improve model fit ($p$ [likelihood ratio test]=0.10).

## Discussion

Among 88,975 individuals followed for a median of 7.1 years, there was a non-linear association between sleep regularity and the risk of mortality; mortality rates were highest in persons with the most irregular sleep and decreased approximately linearly as sleep regularity approached its median, after which the decrease began to plateau. Our findings were independent of several illness (including history of cancer and CVD at baseline), sleep duration, sleep fragmentation, and other confounding factors. Overall, these data indicate a relationship between sleep regularity and longevity in a large community-based cohort.

Physiological processes associated with CVD and cancer are under circadian control. Mutations or deletions to circadian clock genes such as *CLOCK, PER, and BMAL1* influence BP, endothelial function, and glucose homeostasis (*Viswambharan et al., 2007*; *Curtis et al., 2007*; *Anea et al., 2009*;

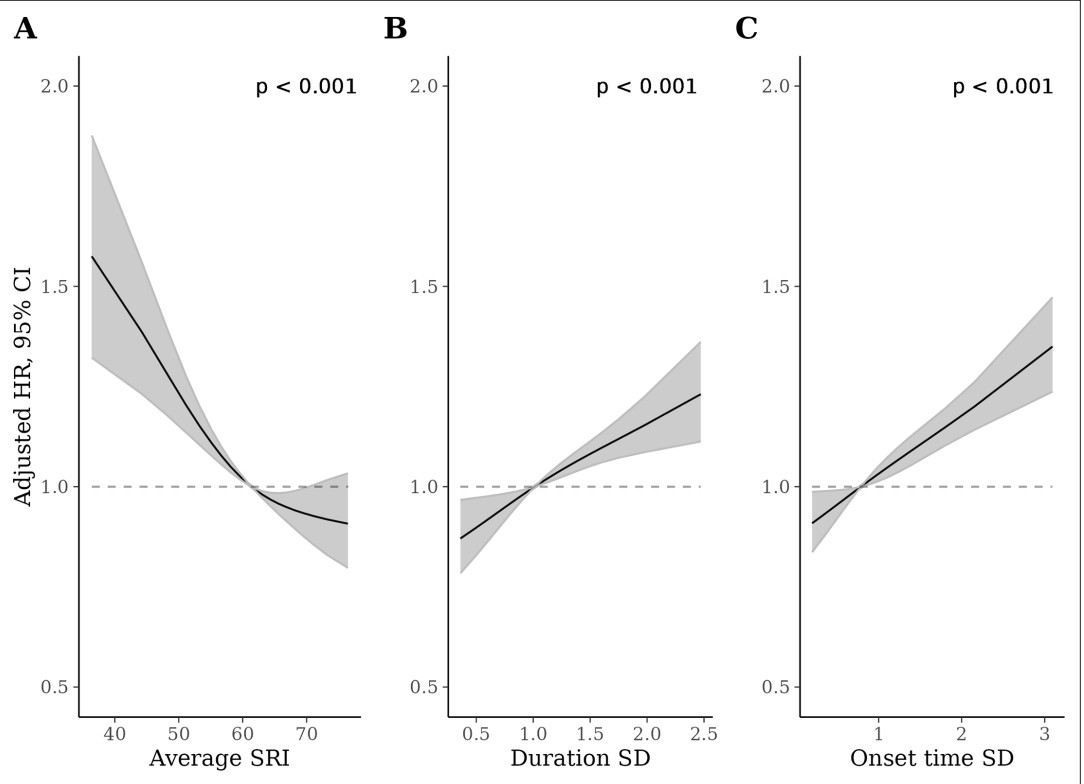

**Figure 3.** Adjusted hazard ratios (HRs) for all-cause mortality for the sleep regularity index (SRI) (**A**), sleep duration standard deviation (SD) (**B**), and sleep onset SD (**C**) measures. p-Values from global (2 degrees of freedom) test of exposure spline term. HRs are relative to the median SRI (SRI = 60). HRs were estimated using Cox proportional hazards models, adjusted for age, Townsend deprivation index, sex, antidepressant, antipsychotic, and sedative medication, ethnicity, household income, education, smoking status (former, current, never), smoking pack years, shift work, retirement status, and sick or disabled (self-reported employment category). All continuous confounders and the sleep regularity metrics were modeled with restricted cubic splines (knots at 10th, 50th, and 90th percentiles) to allow for departures from linearity.

*Rudic et al., 2004*). Both major (e.g., chronic shift work) and minor (e.g., daylight savings transitions) stressors to the circadian system have been associated with a higher risk of CVD (*Torquati et al., 2018*; *Janszky and Ljung, 2008*). Similarly, circadian misalignment has also been implicated in the pathogenesis of cancer. For example, circadian clocks are critical to the orchestration of cell division (*Reddy et al., 2005*), and altered clock function can precipitate aberrant cell proliferation (*Wang et al., 2016*) as well as growth and DNA damage in cancer cells (*Gery et al., 2006*). Many systems are under circadian influence, including the sleep-wake cycle, and less clear has been the extent to which differences in sleep regularity are related to negative health outcomes. We extend this research by demonstrating that differences in sleep regularity are associated with the risk of mortality from both CVD and cancer.

Whereas sleep regularity has not been examined with respect to incident cancer or mortality, the current findings extend research showing that greater sleep-wake variability, as measured by the SD of sleep onset or duration, was independently associated with a higher risk of incident CVD in the multi-ethnic study of atherosclerosis but not the UKB (*Nikbakhtian et al., 2021*; *Huang et al., 2020*) We demonstrate that the SRI contains information about mortality risk beyond that contained in the SD of sleep duration and onset, whereas the converse was not the case. The SRI may be superior to the SD-based metrics because the SRI captures rapid changes in sleep patterns across consecutive days, as compared to the SD-based metrics which only calculate deviation from an individual's average. Rapid changes in sleep timing have been hypothesized as being principally challenging for the circadian system to accommodate (*Phillips et al., 2017*) which may, in turn, produce negative health outcomes.

We found evidence that hazard rates across levels of SRI were non-proportional (i.e., varied across the follow-up period) for all-cause and cancer mortality (which accounted for most deaths), though not for CVD mortality. For cancer mortality, HRs for low SRI compared to the median were largest in the earliest follow-up period and decreased thereafter. One plausible interpretation of this finding is that irregular sleep may be a manifestation of the underlying physiological processes of cancer itself or of cancer treatment (i.e., the SRI-cancer mortality association may be due to reverse causation). However, this thesis is challenged by the fact that associations between the SRI and cancer mortality remained similar after adjusting for history of cancer at baseline. In the case of CVD mortality, no such evidence of a decline in HRs over follow-up time was evident; a potential causal role of irregular sleep on CVD death cannot be easily ruled out.

Sleep of insufficient or excessive duration is associated with many adverse health outcomes, including increased mortality risk (*Cappuccio et al., 2010*). As of 2022, sleep duration was included by the American Heart Association in their Essential Eight guidelines for CVD prevention (*Lloyd-Jones et al., 2022*). However, sleep is far more complex than its habitual duration and quality, with sleep regularity receiving comparatively little attention. As sleep-tracking wearables become more accessible, objective measurement of sleep regularity has the potential for public and clinical use. Much like sleep duration, replicating the current findings across different samples will be necessary for establishing population norms and clinical targets. Furthermore, identifying the determinants of poor sleep regularity may be of import, not only considering biological factors, but broader social determinants that impact circadian rhythmicity (e.g., racial/ethnic disparities [*Chung et al., 2021*], neighborhood factors [*Richardson et al., 2021*]) and consequently overall health.

Our study is not without limitations. First, the study was observational. We are, therefore, unable to establish cause and effect. Although we performed extensive analyses to control for confounding, we cannot exclude the possibility that our results are explained by residual confounding. As such, although therapies exist for improving sleep regularity, it's not clear if these interventions are able to extend the lifespan. Second, sleep and wake were estimated through activity patterns from accelerometry. As compared to polysomnography, there is the potential to misclassify sleep and wake, although accelerometry is more suited to estimate circadian patterns over several days; there are several strengths to using accelerometry (e.g., days of continuous recording, minimal technical apparatus affecting sleep quality), making it the recommended clinical tool for assessing circadian rhythms (*Smith et al., 2018*). In addition, sleep diaries in the UKB were not available. Consequently, the algorithm we used to determine sleep and wake relied on the identification of a main 'sleep period time window' and did not identify napping.

Circadian rhythms have a major influence on health and disease. Although sleep wake timing is under circadian control, research on sleep regularity as a risk factor for mortality was equivocal. These data suggest sleep regularity as an important correlate of longevity, independent of sleep duration, fragmentation, and quality. Future work is needed to determine the underlying mechanisms to inform possible interventions to extend the lifespan.

## Acknowledgements

This research has been conducted using the UK Biobank Resource under project ID 70607. Dr. Pase is supported by a National Health and Medical Research Council of Australia Investigator Grant (GTN2009264) with sleep research funding from the National Health and Medical Research Council of Australia (GTN1158384), National Institute on Aging (R01 AG062531), and Alzheimer's Association (2018-AARG-591358). Dr. Baril is funded by the Banting Fellowship Program (#454104). The funders of the study had no role in study design, data collection, data analysis, data interpretation, or writing of the report.

## Additional information

### Funding

| Funder | Grant reference number | Author |
|---|---|---|
| National Health and Medical Research Council | GTN2009264 | Matthew P Pase |
| National Health and Medical Research Council | GTN1158384 | Matthew P Pase |
| National Institute on Aging | AG062531 | Matthew P Pase |
| Alzheimer's Association | 2018-AARG-591358 | Matthew P Pase |
| Banting Research Foundation | #454104 | Andree-Ann Baril |

The funders had no role in study design, data collection and interpretation, or the decision to submit the work for publication.

### Author contributions

Lachlan Cribb, Data curation, Formal analysis, Methodology, Writing – original draft, Writing – review and editing; Ramon Sha, Writing – original draft, Writing – review and editing; Stephanie Yiallourou, Supervision, Methodology, Writing – original draft, Writing – review and editing; Natalie A Grima, Andree-Ann Baril, Methodology, Writing – review and editing; Marina Cavuoto, Data curation, Writing – review and editing; Matthew P Pase, Conceptualization, Supervision, Funding acquisition, Project administration, Writing – review and editing

### Author ORCIDs

Lachlan Cribb ⬡ https://orcid.org/0000-0001-5276-5249
Stephanie Yiallourou ⬡ http://orcid.org/0000-0002-2433-6372
Matthew P Pase ⬡ https://orcid.org/0000-0002-4143-8485

Reviewer #1 (Public Review): https://doi.org/10.7554/eLife.88359.3.sa1
Reviewer #2 (Public Review): https://doi.org/10.7554/eLife.88359.3.sa2
Author Response https://doi.org/10.7554/eLife.88359.3.sa3

## Additional files

### Supplementary files

• MDAR checklist

### Data availability

Data from the UK Biobank are available, pending application approval from: https://www.ukbiobank.ac.uk/. All analysis code is publicly available at: https://osf.io/tf5xu/ under a CC-By Attribution 4.0 International license.

The following dataset was generated:

| Author(s) | Year | Dataset title | Dataset URL | Database and Identifier |
|---|---|---|---|---|
| Cribb L, Pase M | 2023 | Sleep regularity and mortality | https://osf.io/tf5xu/ | Open Science Framework, tf5xu |

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

## Appendix 1

### Methods: Removal of low-quality accelerometer data

Accelerometry data of low quality were removed using established UKB criteria;

incongruity of self-reported wear time and accelerometer wear time data (5%); insufficient wear time (<72 hr; 5%); and poorly calibrated data (<1%). Last, data were removed for participants in which GGIR was unable to determine a sleep window (5%) and for participants providing less than two valid SRI measurements (i.e., 2 24 hr wear periods; <1%). In total, 88,975 (84%) participants provided valid SRI data and were included in the study.

# Appendix 2

**Appendix 2—table 1.** STROBE statement—checklist of items that should be included in reports of cohort studies.

| | Item No | Recommendation |
|---|---|---|
| Title and abstract | 1 | (a) Indicate the study's design with a commonly used term in the title or the abstract (see title) |
| | | (b) Provide in the abstract an informative and balanced summary of what was done and what was found (see pg 2) |
| **Introduction** | | |
| Background/rationale | 2 | Explain the scientific background and rationale for the investigation being reported (pg 3) |
| Objectives | 3 | State-specific objectives, including any prespecified hypotheses (pg 3) |
| **Methods** | | |
| Study design | 4 | Present key elements of study design early in the paper (pg 3) |
| Setting | 5 | Describe the setting, locations, and relevant dates, including periods of recruitment, exposure, follow-up, and data collection (pg 3) |
| Participants | 6 | (a) Give the eligibility criteria, and the sources and methods of selection of participants. Describe methods of follow-up (pg 3–4) |
| | | (b) For matched studies, give matching criteria and number of exposed and unexposed NA |
| Variables | 7 | Clearly define all outcomes, exposures, predictors, potential confounders, and effect modifiers. Give diagnostic criteria, if applicable (pg 4) |
| Data sources/measurement | 8* | For each variable of interest, give sources of data and details of methods of assessment (measurement). Describe comparability of assessment methods if there is more than one group (pg 3–4) |
| Bias | 9 | Describe any efforts to address potential sources of bias (pg 5) |
| Study size | 10 | Explain how the study size was arrived at (pg 3 and appendix) |
| Quantitative variables | 11 | Explain how quantitative variables were handled in the analyses. If applicable, describe which groupings were chosen and why (pg 5) |
| Statistical methods | 12 | (a) Describe all statistical methods, including those used to control for confounding (pg 4–5, *Appendix 3—figure 1*) |
| | | (b) Describe any methods used to examine subgroups and interactions NA |
| | | (c) Explain how missing data were addressed (pg 4–5) |
| | | (d) If applicable, explain how loss to follow-up was addressed |
| | | (e) Describe any sensitivity analyses (pg 5) |
| **Results** | | |
| Participants | 13* | (a) Report numbers of individuals at each stage of study—e.g., numbers potentially eligible, examined for eligibility, confirmed eligible, included in the study, completing follow-up, and analyzed (pg 3) |
| | | (b) Give reasons for non-participation at each stage |
| | | (c) Consider use of a flow diagram. Not considered necessary but can be created upon request |
| Descriptive data | 14* | (a) Give characteristics of study participants (e.g., demographic, clinical, social) and information on exposures and potential confounders (*Table 1*) |
| | | (b) Indicate number of participants with missing data for each variable of interest. Missing data were infrequent, as described in Methods |
| | | (c) Summarize follow-up time (e.g., average and total amount) (pg 5) |
| Outcome data | 15* | Report numbers of outcome events or summary measures over time (pg 5) |

*Appendix 2—table 1 Continued on next page*

*Appendix 2—table 1 Continued*

| | Item No | Recommendation |
|---|---|---|
| Main results | 16 | (a) Give unadjusted estimates and, if applicable, confounder-adjusted estimates and their precision (eg, 95% confidence interval). Make clear which confounders were adjusted for and why they were included (Figures and Appendix figures) |
| | | (b) Report category boundaries when continuous variables were categorized NA |
| | | (c) If relevant, consider translating estimates of relative risk into absolute risk for a meaningful time period (**Figure 2**) |
| Other analyses | 17 | Report other analyses done—e.g., analyses of subgroups and interactions, and sensitivity analyses (pg 6) |
| **Discussion** | | |
| Key results | 18 | Summarize key results with reference to study objectives (pg 7) |
| Limitations | 19 | Discuss limitations of the study, taking into account sources of potential bias or imprecision. Discuss both direction and magnitude of any potential bias (pg 8) |
| Interpretation | 20 | Give a cautious overall interpretation of results considering objectives, limitations, multiplicity of analyses, results from similar studies, and other relevant evidence (pg 8) |
| Generalizability | 21 | Discuss the generalizability (external validity) of the study results (pg 7–8) |
| **Other information** | | |
| Funding | 22 | Give the source of funding and the role of the funders for the present study and, if applicable, for the original study on which the present article is based (pg 8) |

Note: An Explanation and Elaboration article discusses each checklist item and gives methodological background and published examples of transparent reporting. The STROBE checklist is best used in conjunction with this article (freely available on the Web sites of PLoS Medicine at http://www.plosmedicine.org/, Annals of Internal Medicine at http://www.annals.org/, and Epidemiology at http://www.epidem.com/). Information on the STROBE Initiative is available at http://www.strobe-statement.org.

*Give information separately for exposed and unexposed groups.

## Appendix 3

**Appendix 3—table 1.** Correlation between sleep regularity index and standard deviation-based regularity metrics.

| Regularity measure | Sleep regularity index | Sleep duration SD | Sleep onset SD |
| --- | --- | --- | --- |
| Sleep regularity index | 1 | –0.32 | –0.42 |
| Sleep duration SD | –0.32 | 1 | 0.55 |
| Sleep onset SD | –0.42 | 0.55 | 1 |

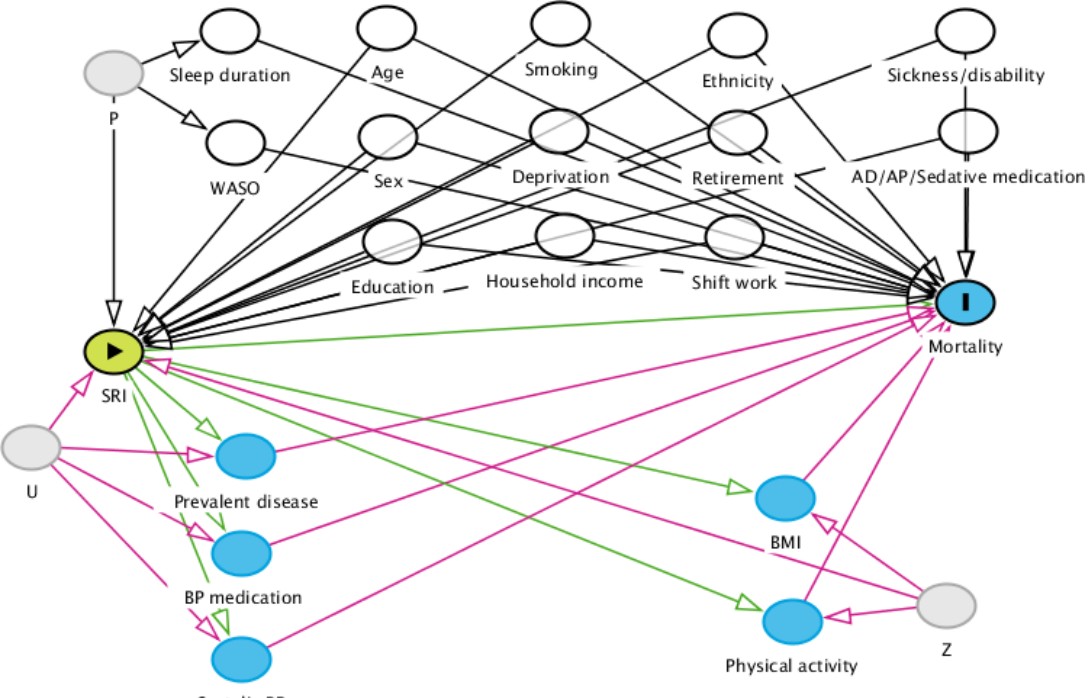

**Appendix 3—figure 1.** Directed acyclic graph for identification of adjustment variables. The green node indicates the exposure variable (*SRI*), and the blue node (*Mortality*) is the outcome variable. Pale gray nodes indicate unobserved variables; white nodes indicate a variable which has been conditioned on (by regression adjustment or restriction). Paths in red are biasing paths. Arrows indicate the direction of causal effect between two nodes. *P* is an unobserved variable representing unmeasured causes of sleep habits (e.g., genetics). *U* is an unobserved variable representing unmeasured causes of disease and cardiovascular dysfunction (e.g., genetics, biological aging). *Z* is an unobserved variable representing unmeasured causes of health behaviors (e.g., personality factors, genetics). Green paths from *SRI* to *Prevalent disease*, *BP medication*, *Systolic BP*, *BMI*, and *Physical activity* and from these nodes to *Mortality* represent potential mediation of an SRI effect. Conversely, red paths indicate potential sources of confounding (e.g., a backdoor path from *Mortality* to *Prevalent disease* to *SRI* via *U*). Given the current evidence base, we are unable to determine whether and to what extent variables such as *Prevalent disease* act as mediators or confounders (via *U*) of the SRI-mortality association. AP = anti-psychotic; AD = antidepressant; BMI = body mass index; BP = blood pressure; CVD = cardiovascular disease; Deprivation = the Townsend deprivation index; SRI = sleep regularity index; WASO = wake after sleep onset.

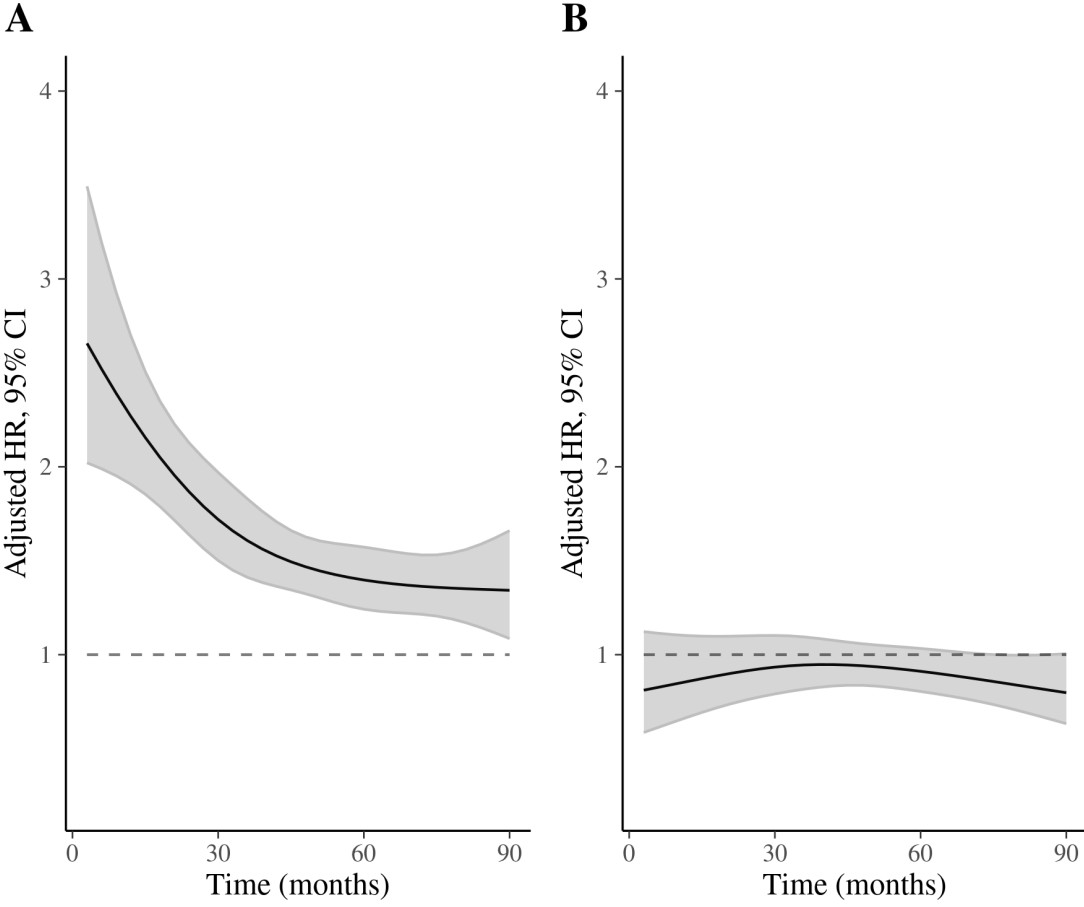

**Appendix 3—figure 2.** Time-varying hazard ratios (HRs) for 5th and 95th percentiles of sleep regularity index (SRI) (relative to median) for all-cause mortality. (**A**) HRs for 5th percentile vs median SRI; (**B**) HRs for 95th percentile vs median SRI. Discrete-time hazards model including time (aggregated into 3-month intervals and modeled with a restricted cubic spline with knots at the 5th, 35th, 65th, and 95th percentiles), SRI, and an SRI by time interaction. Adjusted for age, Townsend deprivation index, sex, antidepressant, antipsychotic, and sedative medication, ethnicity, household income, education, smoking status (former, current, never), smoking pack years, shift work, retirement status, and sick or disabled (self-reported employment category). All continuous confounders and the SRI were modeled with restricted cubic splines (knots at 10th, 50th, and 90th percentiles) to allow for departures from linearity. There was strong evidence of an interaction between time and SRI (p [interaction]<0.001).

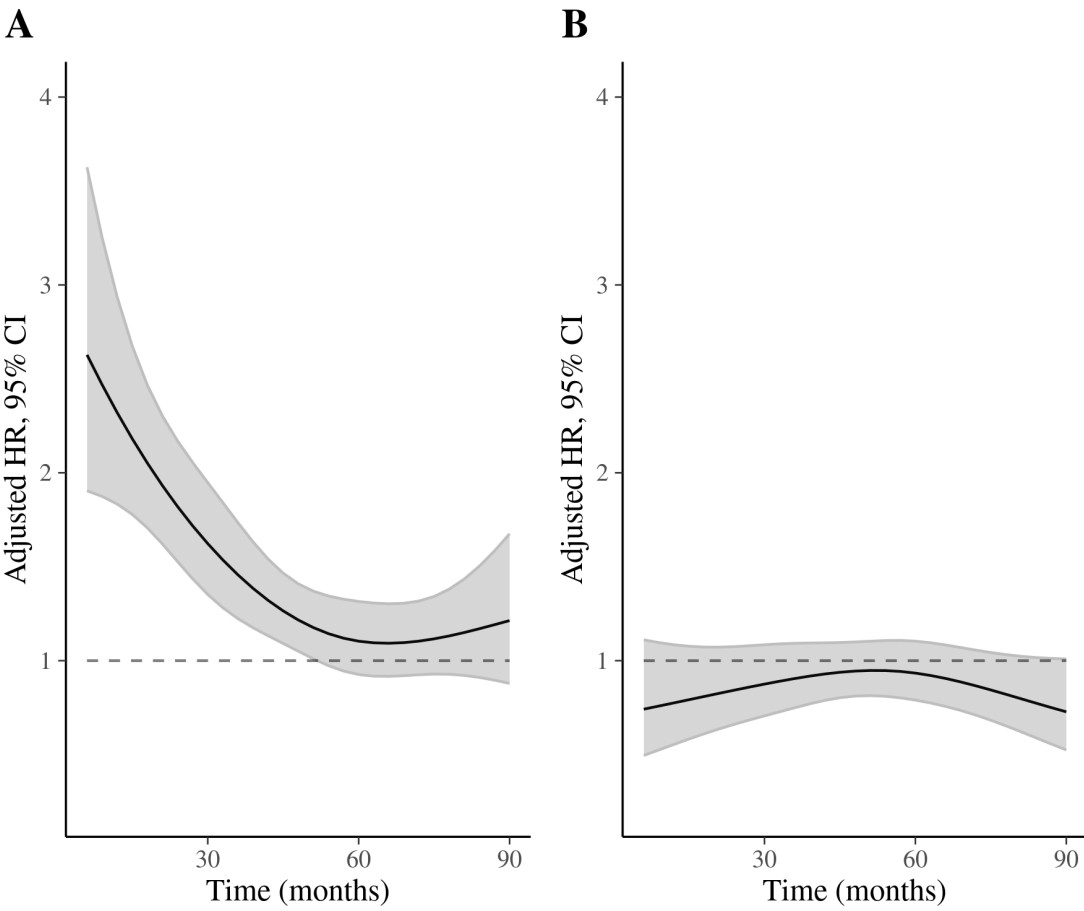

**Appendix 3—figure 3.** Time-varying hazard ratios (HRs) for 5th and 95th percentiles of sleep regularity index (SRI) (relative to median) for cancer mortality. (**A**) Hazard ratios for 5th percentile vs median SRI; (**B**) HRs for 95th percentile vs median SRI. Discrete-time hazards model including time (aggregated into 3-month intervals and modeled with a restricted cubic spline with knots at the 5th, 35th, 65th, and 95th percentiles), SRI, and an SRI by time interaction. Adjusted for age, Townsend deprivation index, sex, antidepressant, antipsychotic, and sedative medication, ethnicity, household income, education, smoking status (former, current, never), smoking pack years, shift work, retirement status, and sick or disabled (self-reported employment category). All continuous confounders and the SRI were modeled with restricted cubic splines (knots at 10th, 50th, and 90th percentiles) to allow for departures from linearity. There was strong evidence of an interaction between time and SRI ($p$ [interaction]<0.001).

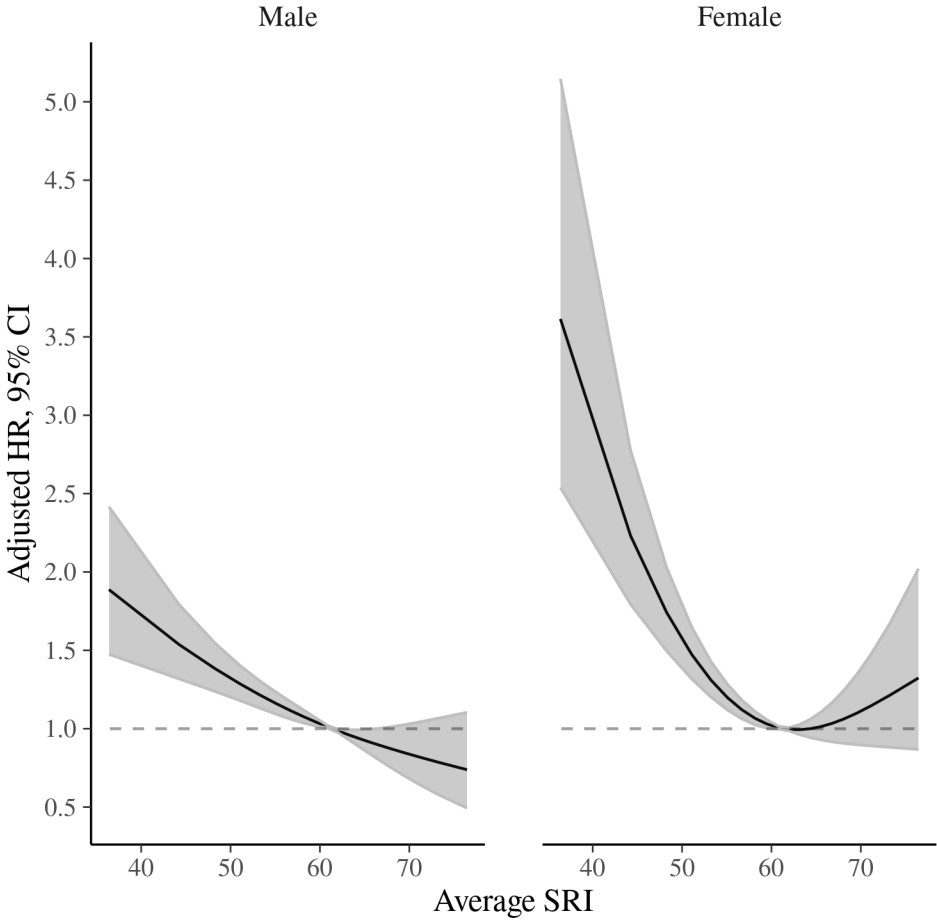

**Appendix 3—figure 4.** Sleep regularity index (SRI) and cardiovascular disease (CVD)-specific mortality by sex. Adjusted for age, Townsend deprivation index, antidepressant, antipsychotic, and sedative medication, ethnicity, household income, education, smoking status (former, current, never), smoking pack years, shift work, retirement status, and sick or disabled (self-reported employment category). Hazard ratios are relative to the median SRI (SRI=60).

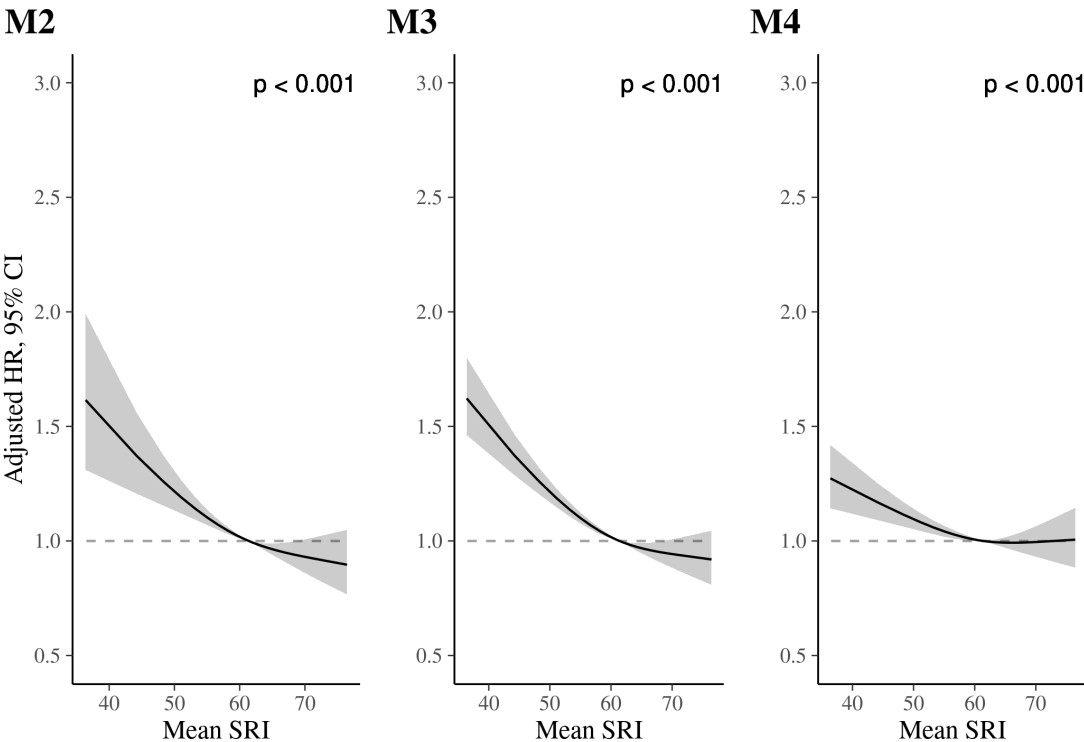

**Appendix 3—figure 5.** Sleep regularity index (SRI) and all-cause mortality in sensitivity analyses. p-Values from global (2 degrees of freedom) test of spline term. Hazard ratios (HRs) are relative to the median SRI (SRI = 60). Model 2 (**M2**) adjustments: Adjusted for age, Townsend deprivation index, sex, antidepressant, antipsychotic, and sedative medication, ethnicity, household income, education, smoking status (former, current, never), smoking pack years, shift work, retirement status, and sick or disabled (self-reported employment category), average sleep time, and average wake after sleep onset time. M2 results: HRs, relative to the median SRI, were 1.42 (95% CI: 1.31, 1.55) and 0.90 (95% CI: 0.80, 1.00) for SRI at the 5th and 95th percentiles, respectively. Model 3 (**M3**) adjustments: Adjusted for age, Townsend deprivation index, sex, antidepressant, antipsychotic, and sedative medication, ethnicity, household income, education, smoking status (former, current, never), smoking pack years, shift work, retirement status, and sick or disabled (self-reported employment category), and history of diabetes, cancer, mental and behavioral disorder, neurological illness, and cardiovascular illness. M3 results: HRs, relative to the median SRI, were 1.46 (95% CI: 1.35, 1.58) and 0.93 (95% CI: 0.83, 1.03) for the 5th and 95th percentiles of SRI, respectively. Model 4 (**M4**) adjustments: Model 3 with additional adjustment for body mass index (BMI), moderate and vigorous physical activity, systolic blood pressure, and blood pressure medication. M4 results: HRs, relative to the median SRI, were 1.20 (95% CI: 1.11, 1.31) and 1.00 (95% CI: 0.90, 1.12) for the 5th and 95th percentiles, respectively.

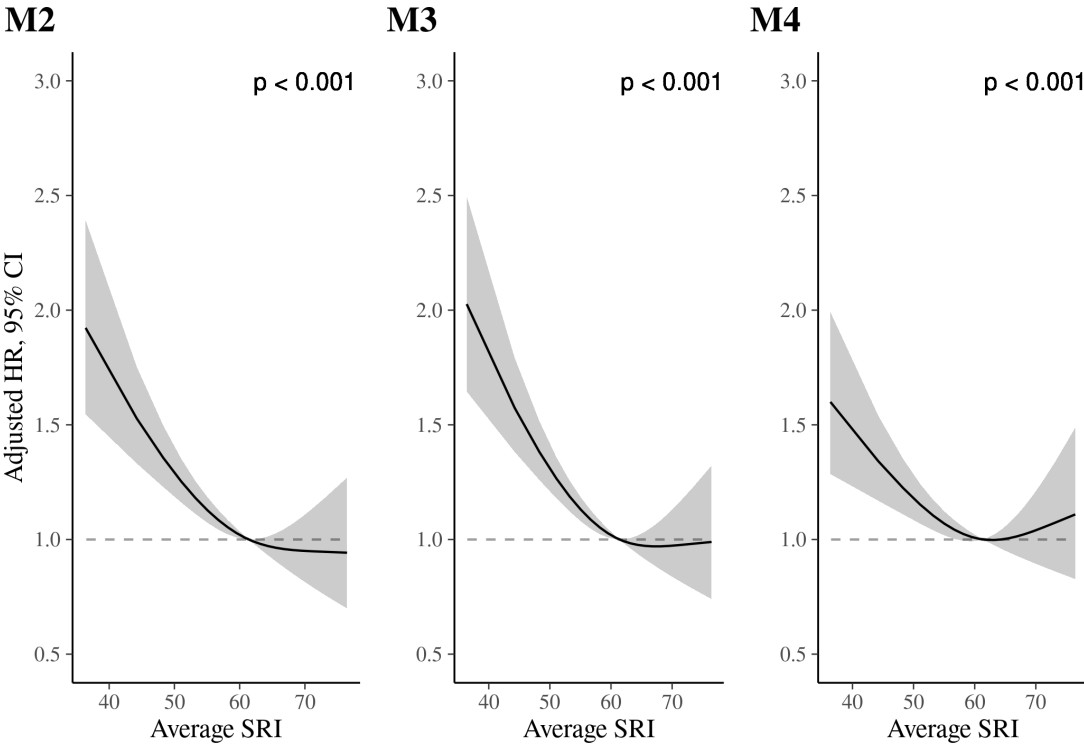

**Appendix 3—figure 6.** Sleep regularity index (SRI) and cardiovascular disease (CVD) mortality in sensitivity analyses. p-Values from global (2 degrees of freedom) test of spline term. Hazard ratios (HRs) are relative to the median SRI (SRI = 60). Model 2 (**M2**) adjustments: Adjusted for age, Townsend deprivation index, sex, antidepressant, antipsychotic, and sedative medication, ethnicity, household income, education, smoking status (former, current, never), smoking pack years, shift work, retirement status, and sick or disabled (self-reported employment category), average sleep time, and average wake after sleep onset time. M2 results: HRs were 1.66 (95% CI: 1.40, 1.96) and 0.95 (95% CI: 0.73, 1.22) for the 5th and 95th percentile vs. the median SRI, respectively. Model (**M3**) adjustments: Adjusted for age, Townsend deprivation index, sex, antidepressant, antipsychotic, and sedative medication, ethnicity, household income, education, smoking status (former, current, never), smoking pack years, shift work, retirement status, and sick or disabled (self-reported employment category), and history of diabetes, cancer, mental and behavioral disorder, neurological illness, and cardiovascular illness. M3 results: HRs were 1.73 (95% CI: 1.47, 2.02) and 0.99 (95% CI: 0.77, 1.26) for the 5th and 95th percentiles, respectively. Model 4 (**M4**) adjustments: Model 3 with additional adjustment for body mass index (BMI), moderate and vigorous physical activity, systolic blood pressure, and blood pressure medication. M4 results: HRs were somewhat attenuated: 1.43 (95% CI: 1.21, 1.69) and 1.09 (95% CI: 0.85, 1.40), for the 5th and 95th percentiles, respectively.

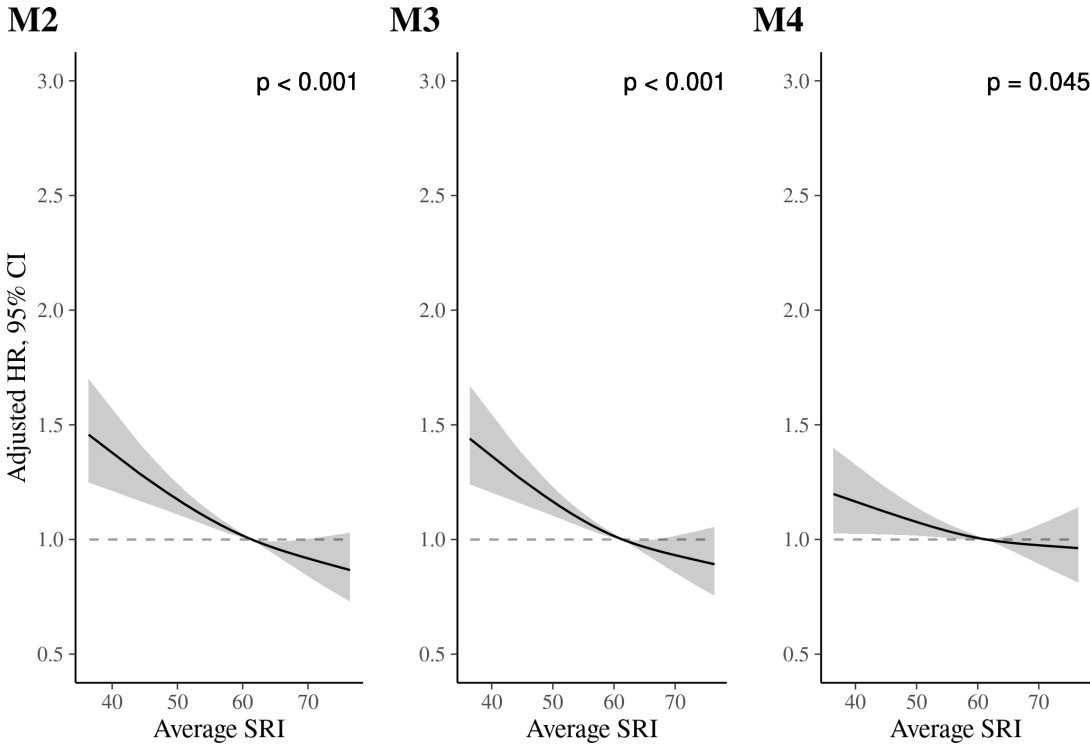

**Appendix 3—figure 7.** Sleep regularity index (SRI) and cancer mortality in sensitivity analyses. p-Values from global (2 degrees of freedom) test of spline term. Hazard ratios (HRs) are relative to the median SRI (SRI = 60). Model 2 (**M2**) adjustments: Adjusted for age, Townsend deprivation index, sex, antidepressant, antipsychotic, and sedative medication, ethnicity, household income, education, smoking status (former, current, never), smoking pack years, shift work, retirement status, and sick or disabled (self-reported employment category), average sleep time, and average wake after sleep onset time. M2 results: HRs were 1.35 (95% CI: 1.20, 1.52) and 0.88 (95% CI: 0.76, 1.02) for the 5th and 95th percentile vs. the median SRI, respectively. Model 3 (**M3**) adjustments: Adjusted for age, Townsend deprivation index, sex, antidepressant, antipsychotic, and sedative medication, ethnicity, household income, education, smoking status (former, current, never), smoking pack years, shift work, retirement status, and sick or disabled (self-reported employment category), and history of diabetes, cancer, mental and behavioral disorder, neurological illness, and cardiovascular illness. M3 results: HRs were 1.33 (95% CI: 1.19, 1.49) and 0.90 (95% CI: 0.78, 1.04) for the 5th and 95th percentiles, respectively. Model 4 (**M4**) adjustments: Model 3 with additional adjustment for body mass index (BMI), moderate and vigorous physical activity, systolic blood pressure, and blood pressure medication. M4 results: HRs were 1.15 (95% CI: 1.02, 1.30) and 0.97 (95% CI: 0.84, 1.12) for the 5th and 95th percentiles, respectively.

