## [Editor Report · eLife assessment]

This manuscript provides **fundamental** findings on the association between sleep regularity and mortality in the UK Biobank, which is a popular topic in recent sleep and circadian research in population-based studies. The study is based on a large accelerometer study with validated follow-up of incident diseases and deaths, and the data quality and large sample size are **convincing** and strengthen the credibility of the conclusion. This will be of wide interest to researchers in the sleep study field, epidemiologists, practicing clinicians and the general public.

---

## [Referee Report · Reviewer #1 (Public Review)]

This manuscript provides important evidence on the association between sleep regularity and mortality in the UK Biobank, which is a popular topic in recent sleep and circadian research in population-based studies. The analysis reported robust associations between sleep irregularity and increased total, CVD and cancer mortality, and provided evidence to support the role of sleep and circadian health in disease progression and longevity in human populations. The Sleep Regularity Index (SRI) used in this study is a novel metric that quantifies the consistency in rest-activity rhythms over consecutive 24 hour periods, thus providing objective assessment of potential circadian disruption. The study is based on a large accelerometer study with validated follow-up of incident diseases and deaths. The data quality and large sample size strengthen the credibility of the conclusion. Overall, the analyses are appropriately done and the manuscript is clearly written.

---

## [Referee Report · Reviewer #2 (Public Review)]

This interesting research commendably revealed irregular sleep-wake patterns are associated with higher mortality risk. However, as authors acknowledged, the analysis can not to accurately identify the cause and effect. In my opinion, the causality is important for this topic, cuz, sleep regularity and health (e.g. chronic disease) were long-term simultaneous status. especially given the participants are older. I suggest that the author could utilize MR analysis to find out for more evidence.

---

## [Author Response]

The following is the authors’ response to the original reviews.

Note to reviewer and editor:

In the previous version of the manuscript, we referred to ‘prevalent’ disease at baseline (e.g., prevalent cardiovascular disease). We have since changed this throughout the manuscript to ‘past or prevalent’ disease. This is a more accurate description as we ascertained diseases which occurred prior to baseline but may have been resolved by the time of the accelerometry study.

Responses to reviewer 1:• I assume that not every participant provided data on all 7 nights. Did the authors exclude those who had fewer number of nights with accelerometer data (e.g., only 2-3 days), as the SRI based on fewer nights may not reliably reflect sleep regularity compared with SRI based all 7 consecutive nights?

It is correct that not every participant provided complete accelerometry data. Most participants (88%) provided complete data. We only included participants who provided at least 2 valid measurements of the SRI (requiring valid data for at least 2 pairs of contiguous 24-hour periods). This is described in the appendix, but we have additionally now added this detail to the main text:

“Most participants (88%) provided complete accelerometry data. Participants with fewer than two valid SRI measurements (i.e., less than 2 contiguous 24-hour wear periods; <1%) were excluded.”

We would also like to note that our statistical analysis accounted, to some extent, for the lower reliability of SRI estimates in those with fewer nights of data. In those with sparse data, their estimated average SRI value would be pulled towards the overall sample average relatively more than for those with complete data. This is a consequence of the ‘partial pooling’ of the linear mixed effects model.

• The primary analysis and results were based on restricted cubic spline models that allow assessment of nonlinearity. This is different from the usual strategy that starts with the simpler linear relationship and further explores potential nonlinear relationships. Did the authors have a strong rationale for a nonlinear dose-response relationship between sleep regularity and mortality, so that the assessment of linear relationships was skipped?

We chose to model the SRI with a restricted cubic spline for two reasons. Firstly, we did expect non-linearity to be present a-priori. Partly this was because other sleep exposures (especially sleep time) have known non-linear relationships with health outcomes. We also thought that it is was plausible that a ‘plateau’ might be present, which we wanted to capture. Secondly, we decided that our primary model should be sufficiently flexible from the outset in order that we did not need to make data-driven adjustments to our model specification (e.g., adding non-linear terms depending on the results of hypothesis tests). This approach we believe to be generally safer as making data-driven changes can undermine the validity of standard errors and p-values.1

• Was the proportional hazards assumption violated in the Cox modeling? Were discrete-time hazard models used to address the violation of the modeling assumption? Please clarify.

Yes, the proportional hazards assumption was violated for all models except for the cardiovascular disease death model. This was the rationale for the use of the discrete time hazards model. They allowed for the inclusion of a flexible time by SRI interaction, allowing the hazard ratio to vary over the follow-up period. We have made this clearer in our revision. The following text has been added to the statistical methods:

“In addition to Cox models, discrete-time hazards models, including an interaction between SRI and time (aggregated into 3-month intervals and modeled with a restricted cubic spline with knots at the 5th, 35th, 65th, and 95th percentiles), were fitted to relax the assumption of proportionality and allow hazard ratios (HRs) to vary over time. The SRI by time interaction in this model provided a test of proportionality (a small p value would indicate strong evidence against the proportional hazards assumption).”

• Please provide correlations between different sleep regularity measures. Although different measures lead to the same conclusion, it is interesting that SRI appeared to provide stronger signals with mortality than the other two SD measures. In addition to what was discussed by the authors, another possibility is that SRI also captures the regularity of napping during the day which is common in older populations.

Thank you for this helpful suggestion. We have added a correlation matrix for the different sleep regularity measures (Table S1). We have additionally added the following text to the Results:

“The SRI was modestly negatively correlated with the sleep duration SD (-0.32) and sleep onset time SD ( 0.42; see correlation matrix in Table S1).”

Regarding napping during the day, the algorithm we used to make determinations of sleep and wake unfortunately is not able to identify napping. This is because, in the absence of a sleep diary, it is very difficult to distinguish napping from inactivity in accelerometry data. The algorithm that we used requires the estimation of a ‘sleep period time window’, defining the period from the beginning to the end of the main sleep bout, in which sleep can be identified. Any sleep outside of this window is treated as inactivity. While some methods have been developed to estimate napping time from accelerometry without a sleep diary, we are not aware of any that are validated for adults using wrist worn accelerometers.

This is something that was not sufficiently clear from the current manuscript. We have had added the following text to ensure this is clear in the revised version.

Methods:

“To distinguish sleep from sustained periods of inactivity without reference to a sleep diary (not available in the UKB), GGIR uses an algorithm to determine a daily ‘sleep period time window’ for each participant.11 This defines the time window between the onset and end of the main daily sleep period, during which periods of sustained inactivity are interpreted as sleep. The algorithm does not, by default, detect bouts of sleep outside of this window and hence is not able to identify naps.”

Discussion:

“In addition, sleep diaries in the UKB were not available. Consequently, the algorithm we used to determine sleep and wake relied on the identification of a main ‘sleep period time window’ and did not identify napping..”

• Table 1 - I would suggest adding additional columns showing the variable distributions across quantiles of the SRI, which can help understand the confounding structure and the covariate associations with SRI.

We agree that this is a good idea and we have adjusted Table 1 accordingly.

• Figure 1 and related supplemental Figures: it would be good to label in the figure the specific HR estimate and 95% CI mentioned in the manuscript.

Thank you for this suggestion. We agree that this would be helpful. After some consideration, we have decided to leave the figures as they are for one primary reason. This is that we want to avoid over-emphasising the 5th and 95th quantiles. As discussed above, we chose to present HRs for these quantiles as they would provide a complement to the Figures which would assist in communication (for some readers, the key results might be easier to glean from these numeric summaries than from the Figures). However, we don’t wish to overemphasise these quantiles when the full ‘dose-response’ function we believe to be of the greatest interest.

• Additional stratified analyses by main sociodemographic factors (age, sex, SES, etc) and sleep variables (sleep duration and sleep quality) would be informative to understand the population heterogeneity in the association between sleep regularity and mortality

Thank you for this suggestion. We have assessed effect modification across a range of key background variables (age, sex, household income, sleep duration, moderate to vigorous physical activity, prevalent CVD, and prevalent cancer). This has been added to the results. Where meaningful evidence of effect modification was noted, we have presented results within strata of the effect modifier.

• Some brief discussion on socioeconomic aspects of sleep is needed (the authors focused on the biological mechanisms underlying the observed association), as emerging evidence suggests that sleep health is not only a biological but also a social construct. For example, a recent study in the US found that sleep regularity is the most important contributor to racial/ethnic disparities in sleep health (see PMID: 34498675).

We agree that sleep health is both a biological and social construct. We have added the following text to the discussion to address this comment:

Discussion:

“Furthermore, identifying the determinants of poor sleep regularity may be of import, not only considering biological factors, but broader social determinants that impact circadian rhythmicity (e.g., racial/ethnic disparities32, neighbourhood factors33) and consequently overall health.”

References

1. Harrell FE. Regression modeling strategies: with applications to linear models, logistic regression, and survival analysis. vol 608. Springer; 2001.